# Corporate social responsibility and employee performance in China's manufacturing sector: Exploring the roles of altruistic values and organizational identification

Kaixian Fu🆔[1], Jirapong Ruanggoon🆔[1]*, Jakkrit Thavorn[2]

**1** Faculty of Management Science, Dhonburi Rajabhat University, Bangkok, Thailand, **2** Department of Organization, Entrepreneurship and Human Resource Management, Thammasat Business School, Thammasat University, Bangkok, Thailand

\* jirapong.r@dru.ac.th

## Abstract

### Background

Few studies exploring the relationship between corporate social responsibility(CSR) and employee performance through Social Identity Theory (SIT) have specifically examined the boundary condition of altruistic values within China's authoritarian cultural context.

### Objectives

This study aims to examine the mechanism of CSR's effect on employee performance in an authoritarian cultural setting, thereby advancing SIT and informing the enhancement of organizational management practices in China.

### Methods

Using a combination of purposive and convenience sampling, a survey was administered to 432 employees from seven manufacturing firms in Sichuan Province in October 2024. All constructs, including perceived CSR, employee performance, organizational identification, and altruistic values, were assessed on 5-point Likert scales. Pearson correlation and linear regression analyses were then conducted to examine the hypothesized relationships.

### Results

Descriptive statistics revealed that the manufacturing employees exhibited high perceived CSR, strong organizational identification, moderate in-role performance, high extra-role performance, and low altruistic values. Moreover, regression analysis confirmed that perceived CSR has strong effects on both in-role ($\beta = 0.44$, $p < 0.01$)

**Published:** Peer Review History: PLOS recognizes the benefits of transparency in the peer review process; therefore, we enable the publication of all of the content of peer review and author responses alongside final, published articles. The editorial history of this article is available here: https://doi.org/10.1371/journal.pone.0339484

**Data availability statement:** All relevant data are publicly available from the figshare repository (https://figshare.com/s/5c295618c5fe18c42e04).

**Funding:** The author(s) received no specific funding for this work.

**Competing interests:** The authors have declared that no competing interests exist.

and extra-role performance ($\beta = 0.40$, p < 0.01), and organizational identification ($\beta = 0.39$, p < 0.01). Organizational identification mediated for approximately 20% of the relationship between perceived CSR and both in-role and extra-role performance, and altruistic values served as a significant negative moderator ($\beta = -0.10$, p < 0.05) between perceived CSR and organizational identification.

## Conclusions

SIT partially explains the effect of CSR on employee performance among Chinese manufacturing employees, revealing organizational identification as a mediator moderated by altruistic values. This insight offers Chinese managers a basis to enhance employee performance in the CSR initiatives process.

## Introduction

Corporate Social Responsibility (CSR) refers to corporate initiatives that proactively address the interests of employees and clients, promote social welfare, and protect the environment during business operations [1]. Today, CSR adoption is driven by a combination of internal motivations and external forces, including government mandates, global best practices, stakeholder pressure, and competitive dynamics [2]. In China, CSR adoption is influenced not only by domestic regulations but also by global standards and increasing consumer awareness. Since China's integration into the global economy, companies —particularly labor-intensive manufacturing firms— have come under greater scrutiny regarding their environmental, social, and governance policies [3]. Given its significant contribution to Chinese economic growth, the manufacturing industry employs a substantial portion of the workforce and operates in contexts where employee welfare, environmental sustainability, and social accountability are increasingly prioritized. The Chinese manufacturing sector thus provides a timely and relevant context for examining how CSR impacts internal organizational processes, especially employee outcomes [4].

Existing research indicates that employees' perceived CSR can influence numerous employee outcomes, including job satisfaction [5], organizational citizenship behavior [6], organizational commitment [7], organizational pride, organizational embeddedness, job behaviors [8], organizational identification [9], and job performance [10]. In summary, the relationship between employees' perceived CSR and various employee outcomes has been examined to varying degrees across different cultural contexts.

To explain the relationship between perceived CSR and employee outcomes, SIT has emerged as one of the most prominent theoretical frameworks. SIT posits that individuals seek to view themselves positively by belonging to valued social groups, perceiving these groups as morally superior and distinct from others [11]. This process is based on three primary mechanisms: social categorization, social identification, and social comparison. Social categorization is the process by which individuals define themselves and others based on salient group memberships, such

as job roles, organizational affiliation, or shared values. Social identification occurs when individuals adopt the norms, values, and goals of their group, fostering a sense of belonging and pride [12,13]. Social comparison enables individuals to evaluate the relative status and moral worth of their in-group compared to out-groups, thereby maintaining self-esteem through positive differentiation.

In the workplace, SIT suggests that employees are more inclined to engage in organizational tasks that align with their personal values and promote socially responsible behavior [14]. When companies participate in CSR activities—such as advancing environmental sustainability or volunteering in community development—they position themselves as ethically commendable organizations. Employees who perceive their company as socially responsible tend to develop a stronger sense of organizational identification, aligning the company's ethical stance with their own values [15,16]. This identification enhances intrinsic motivation, commitment, and discretionary effort, all of which are essential for improving organizational performance. Numerous studies have shown that organizational identification mediates the relationship between perceived CSR and positive employee outcomes, including job satisfaction, extra-role behavior, and organizational commitment [3,17,18].

A substantial body of research supports the SIT perspective that perceived CSR influences employee performance. A meta-analysis confirmed that perceived CSR significantly impacts both in-role and extra-role performance [19]. For example, an empirical study of multinational financial corporations revealed that community-oriented CSR positively influenced employees' in-role performance, even when controlling for demographic and organizational factors [20]. In China, perceived CSR was found to be positively associated with job performance among private-sector workers in urban companies [21]. A study showed that positive attitudes toward CSR were linked to increased helping behavior and civic virtue among Indian employees [22]. Another study in China demonstrated that employees who perceive their employer as socially responsible exhibit higher levels of extra-role behaviors, including volunteering for additional tasks and assisting colleagues [23].

Quantitative studies based on SIT revealed that organizational identification could mediate the relationship between CSR and employee outcomes. A study in Western cultural contexts revealed that perceived CSR enhanced organizational identification, which in turn improved employee adjustment and job performance [24]. Organizational identification and job satisfaction were found to serially mediate the link between CSR and work performance among Korean employees [25], while organizational identification and commitment channeled CSR's effects on performance and turnover intention in Saudi workers [26]. Consistent with these findings, a Chinese study found that organizational identification mediated the relationship between perceived CSR and employee behaviors, including turnover intention, helping behavior, and in-role performance [1].

SIT implies the importance of employees' altruistic values [14]. Altruistic values are stable personal beliefs that prioritize the well-being of others and society over self-interest [27]. These values align with the other-regarding orientation of CSR. According to SIT, employees' internalization and interpretation of organizational norms and practices are influenced by their altruistic values. SIT posits that social identification occurs when employees perceive congruence between their personal beliefs and the values endorsed by the organization [28]. Since CSR reflects a commitment to prosocial organizational alignment, employees with strong altruistic values are more likely to identify closely with CSR-oriented organizations due to a sense of moral congruence. Conversely, employees with low altruistic values may view CSR activities as unimportant and disengaging, reducing their likelihood of developing strong organizational identification [29].

Recent research provides empirical support for the notion that altruistic values moderate the relationship between CSR and organizational identification. One study revealed that individuals with a strong moral identity—a construct theoretically comparable to altruism—are more likely to internalize CSR communications and exhibit increased psychological identification with their company [30]. Individuals with high levels of altruism demonstrated significantly greater organizational identification after exposure to CSR campaigns compared to their low-altruism counterparts [31]. Furthermore, employees' prosocial orientation enhanced the impact of perceived CSR on identification and work engagement, underscoring the crucial moderating role of value congruence [32].

Although current research has utilized SIT to explain how CSR influences employee outcomes, a more in-depth investigation is warranted. First, the relationship between perceived CSR and employee performance remains underexplored within the Chinese cultural context, which is characterized by strong employee deference to superiors and a pervasive culture of authority [33,34]. In this context employees' responses to CSR may be driven more by authority rather than by organizational identification. Second, limited research has examined the moderating role of employees' core values, such as altruism, in the relationship between CSR and organizational identification [1]. CSR activities are prosocial and intended to benefit stakeholders beyond the self. Employees who highly value altruism are more likely to internalize the company's CSR initiatives as meaningful and consistent with their personal values [35]. This consistency can strengthen their identification with the company and result in more positive behavioral outcomes. Although prior research has examined moral identity as a moderator between perceived CSR and organizational identification [36], the distinct role of altruistic values within the Chinese organizational context remains underexplored. The primary objective of this study is to further investigate the relationship between perceived CSR and employee performance within the Chinese authoritarian cultural context, as well as to explore the underlying mechanisms. The findings aim to enrich SIT in the Chinese context and improve organizational management effectiveness in the Chinese manufacturing sector.

Based on the preceding literature and logical reasoning, this study contends that the relationship between employees' perceived CSR and their job performance can be robustly explained through SIT, even within China's authoritarian cultural context. This explanatory mechanism is depicted in the theoretical framework (Fig 1).

The framework posits that organizational identification serves as the mediating mechanism through which perceived CSR influences performance. Furthermore, it theorizes that individual differences, such as altruistic values, may moderate this relationship.

As illustrated in Fig 1, this study advances the following three hypotheses:

H1: Perceived CSR has a positive effect on employee performance.

H2: Organizational identification mediates the relationship between perceived CSR and employee performance.

H3: Altruistic values moderate the positive association between perceived CSR and organizational identification.

This study demonstrates that the relationship between perceived CSR and employee performance is partially mediated by organizational identification, and that altruistic values negatively moderate the link between perceived CSR and organizational identification. It enriches SIT within the Chinese cultural context and contributes to improving management practices in Chinese enterprises. The remainder of the study is organized as follows: (1) Methods (sampling and measures); (2) Research results (mediation and moderation analyses); (3) Discussion; (4) Implications; (5) Limitations; (6) Conclusions.

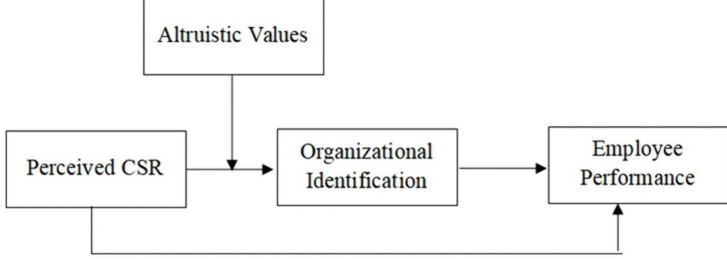

**Fig 1. Theoretical framework.**

## Methods

### Sampling procedure

The study employed a purposive sampling approach to identify eligible enterprises based on predefined criteria: (1) operation in the manufacturing sector, (2) documented implementation of CSR initiatives, and (3) expressed willingness to participate in the survey. Subsequently, a convenience sampling method was used to recruit accessible frontline supervisors—each required to have at least ten direct subordinates and agreed to cooperate—along with their respective teams within these organizations. The sample comprised three state-owned and four privately owned firms from diverse sectors, including machinery, food production, household appliances, and apparel. The employees surveyed were exclusively front-line workers responsible for production tasks, quality control, sales, marketing, and related activities. We contacted the direct supervisors of these frontline employees across the companies and requested their assistance in facilitating the investigation. This approach ensured the confidentiality of the participants [37].

We initially asked 42 supervisors to evaluate their direct subordinates using a 5-point rating scale to assess both in-role and extra-role performance. Subsequently, the supervisors distributed paper-and-pencil questionnaires to their immediate subordinates. The subordinates were asked to rate themselves on perceived CSR, organizational identification, and altruistic values. Each questionnaire was placed in an unsealed envelope, and after completion, the subordinates returned the questionnaires in the same envelopes. Although the process was anonymous, each envelope was discreetly marked to allow supervisors to identify individual submissions. The entire study was conducted from October 5 to October 20, 2024. The day before the formal investigation, we obtained written informed consent from both frontline managers and employees participating in the survey.

A total of 650 pencil-and-paper questionnaires were distributed by 42 supervisors. After eliminating unmatched supervisor-subordinate pairs, the final sample consisted of 432 dyads, resulting in a response rate of 66.5%. The majority of respondents were female (62%), and 68.3% held a bachelor's degree or higher. Most participants were production line factory workers (61.6%), followed by those in sales and marketing (19.2%), quality inspection, maintenance, and other roles. The sample represented five different industries, with the food industry (24.5%) and industrial supplies (23.8%) being the most prominent. Regarding enterprise size, the majority (51.6%) worked in large enterprises, followed by 38.9% in medium enterprises and 9.5% in small enterprises.

### Measurement instruments

The measure of CSR is adapted from a prior study [38]. It encompasses six key items (see S1 Appendix 1): environmental protection, employee welfare, legal compliance, honesty with customers, community engagement, and the promotion of social justice. Sample items are "My employer cares about employee health and safety matters", and "My company strictly adheres to the legal framework". Employees responded on a 5-point rating scale, where "1" signifies " very low" and "5" signifies " very high". A higher mean score on the six-item scale indicates greater employee perceived CSR.

A 5-item scale was used to measure organizational identification [39] (see S1 Appendix 1). Two sample items are: "I consider my organization's successes as my own" and "I talk about my organization by saying 'we' rather than 'they'." Employees responded on a 5-point Likert scale, where "1" signifies " never" and "5" signifies " often ". A higher mean score on the five-item scale indicates stronger employee organizational identification.

We used four items to measure altruistic values: volunteering for a charity, donating clothes or goods, offering a seat on public transportation, and helping neighbors care for others (see S1 Appendix 1). One of the items was: donate clothes or goods to people in need. These items were adapted from previous studies [40–42]. Employees were demanded to respond on a 5-point rating scale, where "1" signifies " never" and "5" signifies " often ". A higher mean score on the four-item scale indicates a greater level of altruistic values.

Employee performance was measured using a six-item scale adapted from two previous studies [43,44] (see S1 Appendix 1). The scale comprises two dimensions: in-role performance and extra-role performance, each containing three items. Two sample items include: "This employee meets the formal performance requirements of the job" (in-role performance) and "This employee volunteers to do tasks for the work group" (extra-role performance). Immediate supervisors rated their subordinates' work performance on these items using a 5-point scale ranging from 1 (never) to 5 (often). Higher mean scores on the six-item scale indicate better employee performance.

To minimize potential common method bias, all measurement items were carefully developed to be clear, simple, and concise, avoiding ambiguous or double-barreled statements. In addition, items for all constructs used specific and unambiguous wording to reduce context-induced mood effects and were further randomized in the questionnaires [45,46].

The present study included four control variables: employees' gender (0 = Male, 1 = Female), tenure, educational level (0 = less than a bachelor's degree, 1 = bachelor's degree or higher), and enterprise size (0 = small enterprise, 1 = medium enterprise, 3 = large enterprise). Dummy variables were set with small enterprise as the baseline.

### Ethical approval statement

This study was approved by Dhonburi Rajabhat University Institutional Review Board; the certificate number is COA NO.006/2567. Informed consent was obtained from all participating employees in this study. This study was also authorized by the corporate human resources manager.

### Data analysis

The data analysis procedure comprised several steps. First, Harman's single-factor test and a single-factor confirmatory measurement model were used to assess common method bias. Subsequently, the measurement model was evaluated by presenting composite reliability (CR) and average variance extracted (AVE) using the Fornell-Larcker criterion, along with the calculation of Cronbach's alpha coefficients. Descriptive statistics and Pearson correlations were then reported. Finally, linear regression analysis was employed to examine the proposed mediating and moderating effects, thereby testing the research hypotheses and the overall theoretical framework.

## Research results

### Common method variance

First, Harman's single-factor test was conducted as an ex-post check [45]. It revealed four factors with eigenvalues exceeding 1, with the first factor explaining 32.72% of the total variance—below the 40% threshold—indicating no significant common method bias in the survey data [47]. Second, confirmatory factor analysis for a single latent factor demonstrated poor model fit (RMSEA = 0.13, CFI = 0.56, TLI = 0.51, SRMR = 0.11). The common factor accounted for only 28.57% of the total variance on average ($R^2$ range: 0.13–0.44), which is well below the 50% threshold, thereby confirming the absence of common method bias.

### Descriptive statistics, correlations, and reliabilities

Table 1 presents means, standard deviations, and correlations between the key variables.

As shown in Table 1, all five constructs demonstrated satisfactory reliability metrics, with composite reliability (CR) values exceeding 0.70, average variance extracted (AVE) indices surpassing 0.50, and Cronbach's alpha coefficients all above 0.70. These results collectively confirm strong measurement reliability for the examined constructs. On a 5-point scale ranging from 1 to 5, composite average scores of 1.81 to 2.61, 2.62 to 3.42, and 3.43 to 4.23 indicate low, moderate, and high levels of the psychological construct, respectively [48]. Thus, employees reported high levels of perceived CSR (M = 4.05), organizational identification (M = 4.11), and low altruistic values (M = 2.45). Supervisors rated employees as demonstrating high extra-role performance (M = 3.79) and moderate in-role performance (M = 3.34).

**Table 1. Descriptive statistics, correlations, and reliabilities (n = 432).**

| | *M* | *SD* | CR | AVE | PCSR | AV | OI | IP | EP |
|---|---|---|---|---|---|---|---|---|---|
| Perceived CSR (PCSR) | 4.05 | 0.69 | 0.81 | 0.55 | *0.80* | | | | |
| Altruistic values (AV) | 2.45 | 0.75 | 0.85 | 0.59 | 0.39** | *0.85* | | | |
| Organizational identification (OI) | 4.11 | 0.68 | 0.84 | 0.51 | 0.42** | 0.36** | *0.84* | | |
| In-role performance (IP) | 3.34 | 1.02 | 0.81 | 0.59 | 0.45** | 0.26** | 0.37** | *0.79* | |
| Extra-role performance (EP) | 3.79 | 0.93 | 0.74 | 0.53 | 0.41** | 0.24** | 0.34** | 0.73** | *0.72* |

Notes: **$p < 0.01$. The italicized numbers on the diagonal represent Cronbach's alpha reliability coefficients. M = mean, SD = standard deviation, CR = composite reliability, AVE = Average Variance Extracted.

According to a prior study, the absolute magnitude of the observed correlation coefficient ranges from 0.10 to 0.39, indicating a weak correlation, and from 0.40 to 0.69, indicating a moderate correlation [49]. Thus, perceived CSR is positively and moderately associated with organizational identification ($r = 0.42$, $p < 0.01$), in-role performance ($r = 0.45$, $p < 0.01$), and extra-role performance ($r = 0.41$, $p < 0.01$). Additionally, organizational identification is positively and weakly associated with extra-role performance ($r = 0.37$, $p < 0.01$) and in-role performance ($r = 0.34$, $p < 0.01$) (Table 1).

## The mediating role of organizational identification in the relationship between perceived CSR and employee performance

The present study employed a three-step regression technique to explore the mediating roles among the variables [50,51]. In Model 1, a regression analysis was conducted to examine the relationship between perceived CSR and employee performance. If this relationship was found to be significant, subsequent steps were performed. In Model 2, organizational identification was regressed on perceived CSR. If organizational identification showed a significant effect, Model 3 was then conducted, in which employee performance was regressed simultaneously on both perceived CSR and organizational identification. Evidence for the mediating role of organizational identification would be supported if it significantly predicted employee performance in this final step.

In all regression analyses, gender, tenure, educational level, and enterprise size were used as control variables. The results of these analyses are presented in Table 2 below.

**Table 2. The relationship between perceived CSR, organizational identification, and employee performance.**

| | Model 1a | Model 1b | Model 2 | Model 3a | Model 3b |
|---|---|---|---|---|---|
| | IP | EP | OI | IP | EP |
| Perceived CSR | 0.44**(9.90) | 0.40**(8.73) | 0.39**(8.69) | 0.36**(7.51) | 0.32**(6.55) |
| Gender | −0.03(−0.72) | −0.04(−0.93) | 0.05(1.11) | −0.04(−0.99) | −0.05(−1.18) |
| Tenure | −0.05(−0.94) | −0.02(−0.42) | −0.07(−1.37) | −0.03(−0.65) | −0.01(−0.15) |
| Educational level | 0.04(0.88) | 0.03(0.60) | −0.05(−1.06) | 0.05(1.14) | 0.04(0.83) |
| Enterprise size | | | | | |
| Medium | 0.04(0.45) | 0.04(0.43) | −0.02(−0.18) | 0.04(0.50) | 0.04(0.48) |
| Large | 0.07(0.84) | 0.08(0.92) | 0.05(0.63) | 0.06(0.71) | 0.07(0.81) |
| Organizational identification | | | | 0.22**(4.73) | 0.21**(4.28) |
| *Adj.$R^2$* | 0.20 | 0.16 | 0.18 | 0.24 | 0.19 |
| *F* | 18.71** | 14.81** | 16.24** | 20.04** | 15.83** |

Notes: Standard coefficients are reported, with t values in parentheses; n = 432. *$p < 0.05$, **$p < 0.01$. The variance inflation factors (VIF) for all variables in the models were below 3. OI = organizational identification, IP = in-role performance, EP = extra-role performance. Homoscedasticity was supported by the plot of standardized residuals against predicted values.

Firstly, Model 1a in Table 2 revealed a significant effect of perceived CSR on in-role performance ($\beta = 0.44$, $p < 0.01$). Additionally, Model 1b showed that perceived CSR had a significant effect on extra-role performance ($\beta = 0.40$, $p < 0.01$). Both effect sizes exceed 0.30, indicating a strong effect based on established thresholds [52]. Therefore, our first hypothesis is strongly supported: perceived CSR is positively associated with employee performance.

Secondly, Model 2 revealed that perceived CSR had a strong predictive effect on organizational identification ($\beta = 0.39$, $p < 0.01$).

Thirdly, when both organizational identification and perceived CSR were included in the model, organizational identification significantly predicted both in-role ($\beta = 0.22$, $p < 0.01$) and extra-role performance ($\beta = 0.21$, $p < 0.01$). Concurrently, the predictive effects of perceived CSR decreased but remained significant, from $\beta = 0.44$ to 0.36 for in-role performance and from $\beta = 0.40$ to 0.32 for extra-role performance ($p < 0.01$ for both; Model 3a and 3b). Therefore, organizational identification serves as a partial mediator in the relationship between perceived CSR and employee performance, providing support for Hypothesis 2.

Fig 2 illustrates the relationships among the variables related to Hypotheses 1 and 2.

Fig 2 demonstrates that organizational identification mediates 20% of the total effect in the relationship between perceived CSR and in-role performance [0.40 × 0.22/0.44]. Similarly, in the relationship between perceived CSR and extra-role performance, organizational identification accounts for 21% of the total effect [0.40 × 0.21/0.40]. Although the proportion of the mediated effect is relatively small, it remains theoretically meaningful and practically significant [53].

Finally, in the present study, none of the control variables—including gender, tenure, educational level, and enterprise size—showed a significant association with either employee performance or organizational identification.

## Altruistic values moderate the relationship between perceived CSR and organizational identification

The present study employed linear regression to test the moderation of altruistic values in the relationship between CSR and organizational identification. To mitigate multicollinearity, perceived CSR and altruistic values were mean-centered prior to analysis [54]. The bootstrap technique involved 2,000 repetitions of sampling. The regression results are presented in Table 3.

As shown in Table 3, the regression analysis revealed a significant but weak negative effect of the interaction term on organizational identification ($\beta = -0.10$, $p < 0.05$) [52]. This finding indicates that altruistic values significantly moderate the relationship between perceived CSR and organizational identification. Therefore, Hypothesis 3 is well supported. To estimate the moderating effect at different levels of altruistic values, the bootstrap technique with 2,000 resamples was employed using PROCESS v4.0. An altruistic values score one standard deviation below the mean was classified as low, while a score one standard deviation above the mean was classified as high. The conditional effects of perceived CSR on organizational identification at these levels of altruistic values are presented in Table 4.

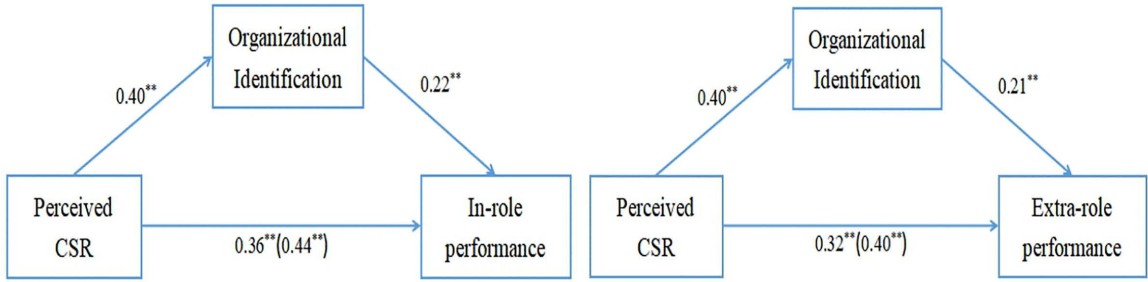

**Fig 2. The mediating role of organizational identification.** Notes: \*\*$p < 0.01$.

**Table 3. Altruistic values moderate the association between perceived CSR and organizational identification.**

| | β | LLCI | ULCI |
|---|---|---|---|
| (Constant) | | 3.81 | 4.29 |
| Perceived CSR | 0.29**(6.12) | 0.18 | 0.39 |
| Altruistic values | 0.26**(5.01) | 0.14 | 0.34 |
| Interaction (Perceived CSR * Altruistic values) | −0.10*(−2.32) | −0.26 | 0.00 |
| Gender | 0.05(1.07) | −0.05 | 0.18 |
| Tenure | 0.04(0.67) | −0.01 | 0.02 |
| Educational level | −0.05(−1.07) | −0.18 | 0.05 |
| Enterprise size | | | |
| Medium | −0.01(−0.10) | −0.25 | 0.21 |
| Large | 0.06(0.71) | −0.15 | 0.30 |
| $F = 16.71^{**}$ | $Adj.R^2 = 0.23$ | | |

Notes: $^*p < 0.05$, $^{**}p < 0.01$. se = standard error. LLCI = Lower Level of Confidence Interval; ULCI = Upper Level of Confidence Interval. The variance inflation factors (VIF) for all variables in the models were below 5. Homoscedasticity was supported by the plot of standardized residuals against predicted values. The variance inflation factors (VIF) for all variables in the models were below 3.

**Table 4. Conditional effects of perceived CSR on organizational identification at values of altruistic values.**

| Altruistic values | Effect | SE | t | LLCI | ULCI |
|---|---|---|---|---|---|
| −0.75 | 0.38 | 0.06 | 6.47** | 0.27 | 0.50 |
| 0.00 | 0.29 | 0.05 | 6.12** | 0.20 | 0.38 |
| 0.75 | 0.19 | 0.07 | 2.94** | 0.06 | 0.32 |

Notes: $^{**}p < 0.01$; SE = standard error; LLCI = Lower Level of Confidence Interval; ULCI = Upper Level of Confidence Interval.

Table 4 illustrates that at lower levels of altruistic values, perceived CSR has a stronger influence on organizational identification (effect = 0.38, 95% CI: 0.27 to 0.50). Conversely, at higher levels of altruistic values, the impact of perceived CSR on organizational identification diminishes (effect = 0.19, 95% CI: 0.06 to 0.32), although this effect remains statistically significant. This indicates that as altruistic values increases, the marginal effect of perceived CSR on employees' organizational identification decreases (see Fig 3).

Furthermore, as shown in Fig 3, after controlling for perceived CSR, employees with high altruistic values exhibit a stronger sense of organizational identification than their less altruistic counterparts. This finding echoes the positive association between altruistic values and organizational identification presented in Table 1.

## Discussion

Drawing on a sample of Chinese manufacturing employees, this study examined their perceived CSR, organizational identification, work performance, and altruistic values, with a focus on the relationships among these variables. The empirical results support the theoretical framework (see Fig 1) and all three research hypotheses.

The current research confirms that manufacturing workers perceive a high level of CSR (M = 4.05, SD = 0.69; Table 1), consistent with a previous study [3], which indicated that Chinese organizations prioritize social responsibility practices. Employees exhibit strong organizational identification (M = 4.11, SD = 0.68; Table 1), suggesting a deep connection with their respective organizations. Furthermore, supervisory ratings indicate moderate levels of in-role performance (M = 3.34,

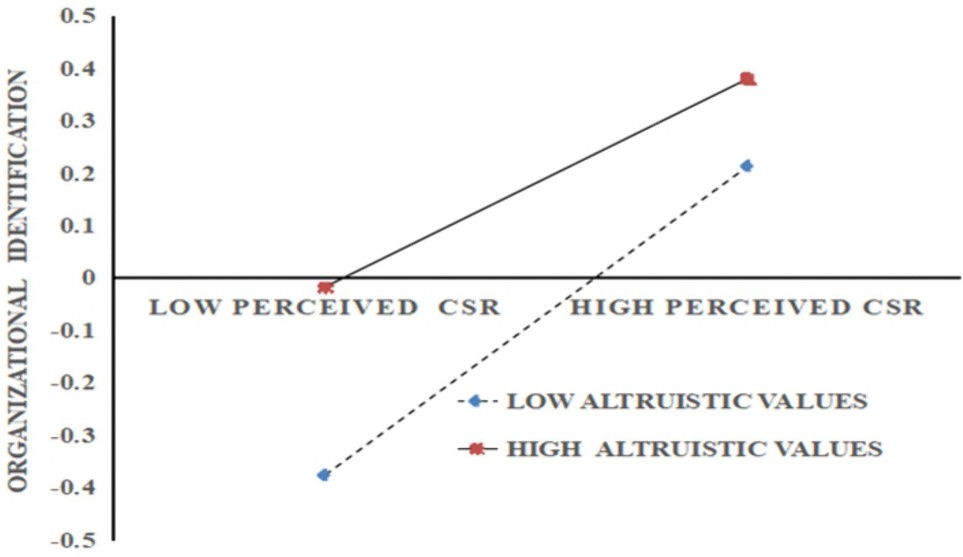

**Fig 3. The moderation of altruistic values.**

SD = 1.02; Table 1), and high levels of extra-role performance (M = 3.79, SD = 0.93; Table 1) among employees, suggesting that frontline manufacturing workers generally demonstrate significant work effort. Lastly, employees report low levels of altruistic values (M = 2.45, SD = 0.75; Table 1), indicating limited concern for the well-being of others.

This study confirms a moderate positive link between perceived CSR and both in-role (r = 0.45) and extra-role (r = 0.41, Table 1) performance, consistent with prior research [55]. For instance, a meta-analysis confirmed that the correlation coefficient between perceived CSR and employee performance generally ranges from 0.33 to 0.45 [56]. Additionally, this study revealed that employees' perceived CSR has a moderately positive correlation with organizational identification (r = 0.42, Table 1), consistent with previous findings [1,25]. For example, an earlier study reported a Pearson correlation of 0.57 between perceived CSR and organizational identification [25]. Furthermore, organizational identification is weakly but positively related to both in-role (r = 0.37) and extra-role (r = 0.34, Table 1) performance, consistent with the findings in a prior study [1].

Regression analysis, controlling for gender, educational level, tenure, and enterprise size, showed that perceived CSR had a significant positive effect on both in-role ($\beta$ = 0.44, $p$ < 0.01) and extra-role performance ($\beta$ = 0.40, $p$ < 0.01; Table 2), thus supporting Hypothesis 1, which posits that perceived CSR is positively related to employee perfor-mance. This conclusion aligns closely with results from previous studies. For instance, one study demonstrated that employees in high-CSR environments outperform their peers [10], while another found that employees' perceived CSR enhances their organizational citizenship behavior [57]. This result not only echoes the findings from studies conducted in Western cultures [58,59], but also shows that CSR has a positive relationship with employee perfor-mance within the context of Chinese culture.

The regression analysis also indicated that perceived CSR has a significant effect on organizational identification ($\beta$ = 0.39, $p$ < 0.01; Model 2 in Table 2). Furthermore, when both organizational identification and perceived CSR were included, organizational identification demonstrated positive effects on both in-role ($\beta$ = 0.22, $p$ < 0.01; Model 3a in Table 2) and extra-role performance ($\beta$ = 0.21, $p$ < 0.01; Model 3b in Table 2). Therefore, Hypothesis 2 is strongly supported, posit-ing that organizational identification mediates the relationship between perceived CSR and employee performance. This finding is consistent with the results of previous research. For example, a prior study demonstrated that organizational

identification mediates the relationships between perceived CSR and turnover intention, in-role job performance, and extra-role performance, as indicated by helping behavior [1,60]. Additionally, a study conducted in China found that organizational identification plays a crucial role in the relationship between employee-perceived CSR and organizational citizenship behavior [6].

The mediating role of organizational identification is effectively explained by SIT. Firstly, according to SIT, individuals are more inclined to affiliate with the organizations that possess a positive social standing and reputation [61], as this affiliation enhances their self-esteem and self-concept [62]. Employees tend to identify more positively with and perceive a sense of oneness with the organizations they regard as socially responsible [63]. Thus, perceived CSR significantly and positively associated with organizational identification among manufacturing employees, as indicated by the regression analysis in the present study (see Model 2 in Table 2).

Secondly, according to SIT, employees who strongly identify with their organizations are more likely to demonstrate favorable attitudes toward their organizations and their responsibilities within them. Compared to employees with lower levels of organizational identification, those with a strong organizational identification are more inclined to exhibit organizational citizenship behaviors [64], prosocial behaviors [65], and knowledge sharing behaviors [66]. Therefore, the regression analysis conducted in this study indicated that organizational identification positively influences both in-role performance and extra-role performance among mamufacturing empoyees (see Model 3a and 3b in Table 2).

Additional analysis indicated that organizational identification served as a partial mediator in this study, accounting for approximately 20% of the effect of perceived CSR on both in-role and extra-role performance. This suggests that about 80% of the total effect of perceived CSR on in-role and extra-role performance remains unexplained by the organizational identification mechanism (see Fig 2). Therefore, other mediating variables warrant further exploration. For example, one study found that organizational trust and job satisfaction can mediate the relationship between perceived CSR and employee performance [66]. Another study demonstrated that work meaningfulness and organizational pride sequentially mediate the link between perceived CSR and job performance among Chinese employees [67]. In addition, another potential explanatory mechanism deserves consideration. Within China's cultural context, where authoritarian norms are prevalent [68], employees' work performance and CSR compliance may result more from deference and loyalty to management than from genuine endorsement of CSR or organizational identification. In summary, beyond the mechanism of organizational identification grounded in SIT, other mechanisms may also partially account for the relationship between perceived CSR and employee performance.

Regression results indicated a significant interaction between perceived CSR and altruistic values on organizational identification ($\beta=-0.10$, $p<0.05$; Table 3), supporting Hypothesis 3 that altruistic values moderate the positive relationship between CSR and identification. The negative moderating effect suggests that, for employees with high levels of altruism, the positive association between perceived CSR and organizational identification is attenuated compared to those with lower altruism. Nonetheless, employees with high altruism tend to exhibit stronger organizational identification than those with low altruism (Table 4 and Fig 3). This finding aligns with a prior study showing that employees' altruistic beliefs moderate the positive relationship between perceived CSR and both organizational commitment and job satisfaction [1]. In light of SIT, employees' values play a crucial role in the process of organizational identification [69]. When organizational values are congruent with employees' values, employees are more likely to develop strong organizational identification [70].

## Implications

This research offers three significant theoretical contributions. First, it extends the cross-cultural applicability of SIT by demonstrating its explanatory power within the Chinese cultural context. While SIT has been extensively applied in Western literature, this study reaffirms that the processes of identification and value congruency remain valid in an authority culture for explaining the impact of CSR on work-related outcomes. Second, this study demonstrates that organizational identification serves as a partial mediator between perceived CSR and employee performance in Chinese culture, indicating that SIT only partially explains this relationship and that

other underlying mechanisms are likely involved. Third, by defining altruistic values as a moderating factor, the study expands the boundary conditions of SIT. Specifically, although employees with high altruism exhibit stronger organizational identification, the relationship between their perceived CSR and organizational identification weakens as altruism increases. This finding enriches the conceptual framework of SIT concerning the relationship between CSR and employee outcomes.

This study offers two practical contributions. First, corporate managers in China's manufacturing sector should recognize the motivational impact of CSR on employees. The data analysis indicates that CSR is significantly and positively correlated with both organizational identification and employee performance. Therefore, managers in the manufacturing industry can intentionally strengthen employees' cognitive and emotional connection to the organization through CSR initiatives, thereby effectively enhancing both their in-role and extra-role performance. Second, managers should focus on cultivating altruistic values among employees, particularly those with lower levels of altruism, as this study reveals that altruistic values negatively moderate the relationship between perceived CSR and organizational identification. Psychological research suggests that interventions such as compassion-based mental training can enhance prosocial behavior and altruistic motivation over time [71].

## Limitations

This study has three notable limitations that warrant attention. First, it employs cross-sectional data, which allows only correlational, not causal, conclusions. Future research should incorporate longitudinal follow-ups with participants to provide more robust evidence supporting the relationships among perceived CSR, organizational identification, and employee performance. Second, the present study examines the associations between perceived CSR and employee performance using an individual-level model, thereby overlooking potential organization-level influences. Future research would benefit from exploring employees' responses to CSR within a multilevel framework. Finally, this study employed a non-probabilistic sampling method, drawing its sample exclusively from manufacturing firms in Sichuan Province. This approach may thereby constrain the generalizability of the findings to other regions or industries. To enhance the external validity of future research, it would be beneficial to employ probability sampling techniques across more diverse geographical contexts.

## Conclusions

This study examined the relationship between perceived CSR, organizational identification, employee performance, and the altruistic values of frontline employees in the Chinese manufacturing sector. The results indicate that perceived CSR is positively associated with employee performance, and this relationship is significantly mediated by organizational identification. Furthermore, the findings demonstrate that altruistic values moderate the relationship between perceived CSR and organizational identification; specifically, as the level of altruistic values increases, the strength of this relationship decreases. This study extends Social Identity Theory and offers valuable insights for enhancing managerial practices in Chinese manufacturing organizations.

## Supporting information

**S1 Appendix. 1 Survey questionnaire used in this study.**
(DOCX)

## Acknowledgments

We would like to express our sincere gratitude to supervisors Pan Baoliang, Wang Xihua, and Yu Liang, among others, for their valuable support.

## Author contributions

**Conceptualization:** Kaixian Fu.

**Investigation:** Kaixian Fu.

**Methodology:** Jirapong Ruanggoon, Jakkrit Thavorn.

**Supervision:** Jirapong Ruanggoon, Jakkrit Thavorn.

**Writing – original draft:** Kaixian Fu.

**Writing – review & editing:** Jirapong Ruanggoon, Jakkrit Thavorn.

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
