## [Decision Letter · Decision Letter 0]

26 Sep 2025

PONE-D-25-38655
Corporate Social Responsibility and Employee Performance in China's Manufacturing Sector: Exploring the Roles of Altruistic Values and Organizational Identification
PLOS ONE

Dear Dr.  Ruanggoon,

Thank you for submitting your manuscript to PLOS ONE. After careful consideration, we feel that it has merit but does not fully meet PLOS ONE’s publication criteria as it currently stands. Therefore, we invite you to submit a revised version of the manuscript that addresses the points raised during the review process.

We look forward to receiving your revised manuscript.

Kind regards,

Federico Zilia

Academic Editor

PLOS ONE

Journal Requirements:

3. Thank you for uploading your study's underlying data set. Unfortunately, the repository you have noted in your Data Availability statement does not qualify as an acceptable data repository according to PLOS's standards.

Additional Editor Comments:

Both reviewers raised important points concerning the clarity of your introduction and literature review, the completeness and transparency of your methodological section (particularly the sampling procedure and validation of constructs), and the rigor of your results interpretation (e.g., more robust tests for mediation, clearer regression assumptions). You are also encouraged to improve figure captions, refine the language for clarity, and expand your discussion to better reflect limitations and alternative explanations.

We look forward to receiving a thoroughly revised manuscript that addresses these points in detail.

Reviewer's Responses to Questions

**Comments to the Author**

1. Is the manuscript technically sound, and do the data support the conclusions?

Reviewer #1: Yes

Reviewer #2: Partly

2. Has the statistical analysis been performed appropriately and rigorously?

Reviewer #1: Yes

Reviewer #2: Yes

3. Have the authors made all data underlying the findings in their manuscript fully available?

Reviewer #1: Yes

Reviewer #2: Yes

4. Is the manuscript presented in an intelligible fashion and written in standard English?

Reviewer #1: Yes

Reviewer #2: No

5. Review Comments to the Author

Reviewer #1: Dear Authors,

I have carefully reviewed your manuscript and appreciate the relevance of the topic, the clarity of your research question, and the soundness of your methodological framework. The manuscript is generally well-written and structured, and the dataset employed has potential to contribute significantly to the literature in the field. Nevertheless, there are several points that could be improved to enhance clarity, transparency, and overall impact.

1.Introduction – Streamlining and Emphasis on Novelty

The introduction provides a solid contextual background and clearly states the research hypotheses. However, certain literature review paragraphs are somewhat redundant or too detailed relative to the focus of the study. Condensing these sections would help direct the reader more quickly to the novelty of your approach.

2. Methods – Completeness of Description

The methodological section is clear but could benefit from a fuller description of:

• The rationale for selecting specific variable transformations and the criteria used for model selection.

• The procedures adopted to check statistical assumptions (e.g., normality, homoscedasticity, absence of multicollinearity (VIFs are reported only in a note to Table 2).

Including these details will improve reproducibility and transparency.

3. Results – Clarity and Integration with Hypotheses

Results are presented in a logical order, but in some cases the link between numerical outputs and the corresponding hypotheses could be made more explicit in the text. This would strengthen the narrative flow and help the reader follow how the evidence addresses each hypothesis.

Additionally, while figures are generally clear, in Figure 1 there is a typographical error (perceived not perceive CSR) and Figure 3 have axes or legends that are difficult to interpret without referring to the text. Also captions should be more self-contained.

4. Discussion – Critical Engagement

The discussion appropriately summarizes the main findings, but could be strengthened by:

• Providing a more explicit reflection on potential alternative explanations for the observed patterns.

• Expanding the implications of limitations, particularly in terms of the dataset’s spatial and temporal scope.

• Ensuring that policy or practical implications are clearly framed as either evidence-based or speculative, to avoid overstatement.

5. Language and Formatting

The manuscript is overall well-written, but some sentences are long and could be simplified for clarity. Minor grammatical issues and small inconsistencies in reference formatting should be addressed with a final proofreading.

In conclusion, this is a well-conceived study that addresses an important question. By streamlining the introduction, enriching methodological details, ensuring tighter integration between results and hypotheses, correcting minor errors (including the typo in Figure 1), and expanding the critical discussion, the manuscript will be significantly strengthened and more compelling to the readership.

Reviewer #2: The paper exploits Social Identity Theory (SIT) to investigate how Corporate Social Responsibility (CSR) impacts employee performance when considering organizational identification as a mediator and altruistic values as a moderator.

However, in order to evaluate the presented results, further clarification and analysis are necessary. Furthermore, not all sections of the manuscript are written with sufficient detail to be accessible to non-specialists.

My overall assessment is that this paper shows potential but some of my remarks should be considered (with related adjustments) in order to proceed with the publication process.

Below, I provide a number of major and minor remarks.

INTRODUCTION

The introduction positions the analysis consistently within the relevant literature but some adjustments are necessary to make the manuscript more intelligible.

The introduction should cover both the theoretical foundations and the related empirical contributions. In addition, it should provide a brief explanation of the objectives of the study and the overall structure of the paper. Ideally, one should be able to grasp the essence of the work by reading only the introduction and the conclusions. This is not the case here and I would therefore recommend including at least a summary of the main findings as well as a clearer description of the paper’s structure.

Other minor issues:

- “Of the numerous theoretical models employed to explain these relations, Social Identity Theory (SIT)” You are not actually referring to a formal economic model. I would rather describe this as a theoretical framework. Similarly, I would avoid referring to Figure 1 as a theoretical model.

- “Furthermore, this research contributes theoretical insights to the CSR literature by illuminating the processes that connect external CSR perceptions with internal value-based responses.” Not very clear.

METHODS

The sampling procedure is not explained. How were the enterprises selected? Without this information the results may not be generalizable to the target population. This clarification is essential in order to properly assess the validity of the manuscript’s findings.

- It would also be useful to provide a description of the measurement items, even if only in the annexes.

- The expression “deprived from a previous study” should be revised, as it appears to be incorrect.

- The subsection “Data Analysis” is too brief. Key information should be presented here rather than scattered across the results section, otherwise the subsection loses its purpose.

RESEARCH RESULT

Although the constructs show good measurement reliability, additional diagnostics are required before the results can be properly evaluated, particularly in light of the detected correlations among the constructs.

Major points:

- Harman’s single-factor test is an ex-post statistical check, not a method to anticipate common method bias. Moreover, this test is limited, as it does not account for the possibility that common method variance (CMV) may load onto multiple general factors. A more appropriate approach would be to conduct a Confirmatory Factor Analysis (CFA) including all items. In this specification an additional latent factor is introduced onto which all items load, in addition to their respective constructs. If this additional factor explains a substantial share of the variance (commonly more than 30%) common method bias is present. The decision to avoid using CFA should therefore be explained and properly justified.

Minor points:

- The regression assumptions should be explicitly verified. If this was done, it should at least be stated that they were confirmed (e.g., multicollinearity and homoscedasticity).

- In Table 2, the analytical steps are correct, but in the text it is reported that: “In Model 2, another independent variable, M, was included in the regression model alongside X as predictors.” Here, however, the mediator is the outcome. The steps should be explained with greater clarity in the text. It is not necessary to follow the variable nomenclature used by Namazi and Namazi (2016); it would be clearer to adopt the nomenclature actually employed in the analysis.

- An increase in the R^2 is not sufficient to establish a mediation effect.

- Why was a bootstrapping method not employed for the mediation analysis as well?

- The reasons why perceived CSR and altruistic values were mean-centered prior to analysis should be mentioned.

- Why did you not consider the SEM framework for your analysis?

DISCUSSION, IMPLICATIONS, LIMITATIONS AND CONCLUSIONS

These sections should be reconsidered in light of the requested revisions. At present, the claims are not, in the reviewer’s view, sufficiently well supported.

6. PLOS authors have the option to publish the peer review history of their article (what does this mean?). If published, this will include your full peer review and any attached files.

Reviewer #1: No

Reviewer #2: No

---

## [Author Response · Author response to Decision Letter 1]

9 Oct 2025

Dear Editor and reviewers

I sincerely appreciate your constructive feedback. The manuscript has been carefully revised with my supervisor, and I believe it now meets the journal's standards. Kindly review the updated version and advise if additional changes are needed.

I have made major revisions to the manuscript, and I will strive to highlight the key changes.

Best regards

To editor

My reply:

Yes, I have made the revisions in accordance with the journal's requirements.

My reply:

Yes, I have carefully completed the PLOS Questionnaire on Inclusivity in Global Research. In addition, I have included a statement regarding the S1 File in the "Ethical Approval Statement" section of the manuscript, as shown below (see blue text):

Ethical Approval Statement

This study was approved by Dhonburi Rajabhat University Institutional Review Board; the certificate number is COA NO.006/2567. Informed consent was obtained from all participating employees in this study. This study was also authorized by the corporate human resources manager. In addition, A completed Questionnaire on Inclusivity in Global Research is available as Supporting Information S1 File.

3. Thank you for uploading your study's underlying data set. Unfortunately, the repository you have noted in your Data Availability statement does not qualify as an acceptable data repository according to PLOS's standards.

My reply:

I have uploaded the raw data for this manuscript to Figshare. The data download link is as follows:

https://figshare.com/s/5c295618c5fe18c42e04

My reply:

I have uploaded the data to Figshare and added a "Data Availability Statement" in the "Introduction" section of the revised manuscript, as shown in the blue text：

Data Analysis

The study first employed Harman's single-factor test and a single-factor confirmatory measurement model to assess common method bias. Subsequently, the measurement model was evaluated by presenting composite reliability (CR) and average variance extracted (AVE) using the Fornell-Larcker criterion, alongside the calculation of Cronbach's alpha coefficients. Descriptive statistics and Pearson correlations were then reported. Finally, linear regression analysis was utilized to test the theoretical framework and research hypotheses.

Data Availability Statement

All data underlying the findings described in this manuscript are fully available without restriction through Figshare at https://figshare.com/s/5c295618c5fe18c42e04

My reply:

Noted. Thank you for the reminder!

Additional Editor Comments:

Both reviewers raised important points concerning the clarity of your introduction and literature review, the completeness and transparency of your methodological section (particularly the sampling procedure and validation of constructs), and the rigor of your results interpretation (e.g., more robust tests for mediation, clearer regression assumptions). You are also encouraged to improve figure captions, refine the language for clarity, and expand your discussion to better reflect limitations and alternative explanations.

My reply:

I have performed major revisions throughout the entire manuscript and have also polished the English language. The revised version presents a clearer line of reasoning, a more logical flow in the introduction, enhanced clarity in the methods section, and a more rigorous presentation of the research findings. In summary, the overall quality of the paper has been significantly improved. Should any further revisions be required, please do not hesitate to let me know.

Once again, I would like to express my sincere gratitude for the editor's comments and suggestions.

To reviewer 1,

Thank you very much for your thoughtful and constructive comments. I have carefully reviewed all of your suggestions, which have significantly improved both the quality of this manuscript and my academic writing skills. I sincerely appreciate your time and expertise.

In response to your feedback, I have thoroughly revised the manuscript. I would be most grateful if you could review these changes and provide any additional guidance you may have.

Once again, please accept my deepest gratitude for your valuable contribution to this work.

Review Comments to the Author

1.Introduction – Streamlining and Emphasis on Novelty

The introduction provides a solid contextual background and clearly states the research hypotheses. However, certain literature review paragraphs are somewhat redundant or too detailed relative to the focus of the study. Condensing these sections would help direct the reader more quickly to the novelty of your approach.

My reply:

I have made major revisions to the "Introduction" section. I have streamlined this part by presenting the research background, theoretical foundation, and limitations of previous studies in a step-by-step manner, making the logic clearer and more fluid. The revised structure is as follows:

“CSR in China→the associations between CSR and employee outcomes→SIT can explain these associations→the content of SIT→literature evidence SIT notions on the link between perceived CST and employee performance→SIT implies the role of altruistic values→prior studies evidence the moderation of altruistic values→the gaps of the previous literature→the aim of the study →theoretical framework and hypotheses”

Due to the extensive revisions made to the "Introduction" section, the changes are too extensive to list individually. Please refer to the revised manuscript to see the updated version.

2. Methods – Completeness of Description

The methodological section is clear but could benefit from a fuller description of:

（1） The rationale for selecting specific variable transformations and the criteria used for model selection.

（2）The procedures adopted to check statistical assumptions (e.g., normality, homoscedasticity, absence of multicollinearity (VIFs are reported only in a note to Table 2).

Including these details will improve reproducibility and transparency.

My reply:

（1）I have added the rationale for selecting specific variable transformations as follows (see the text in blue):

Altruistic Values Moderate the Relationship between Perceived CSR and Organizational Identification

The present paper employed linear regression to test the moderation of altruistic values in the relationship between CSR and organizational identification. To mitigate multicollinearity, perceived CSR and altruistic values were mean-centered prior to analysis[52]. The bootstrap technique involved 2,000 repetitions of sampling. The regression results are presented in Table 3.

（2）I have added the procedures to check statistical assumptions (e.g., normality, homoscedasticity, absence of multicollinearity) to Table 2 (see the text in blue).

Table 2 The relationship between perceived CSR, organizational identification, and employee performance

Model 1a Model 1b Model 2 Model 3a Model 3b

IP EP OI IP EP

Perceived CSR 0.44**(9.90) 0.40**(8.73) 0.39**(8.69) 0.36**(7.51) 0.32**(6.55)

Gender -0.03(-0.72) -0.04(-0.93) 0.05(1.11) -0.04(-0.99) -0.05(-1.18)

Tenure -0.05(-0.94) -0.02(-0.42) -0.07(-1.37) -0.03(-0.65) -0.01(-0.15)

Education level 0.04(0.88) 0.03(0.60) -0.05(-1.06) 0.05(1.14) 0.04(0.83)

Enterprise size

Medium 0.04(0.45) 0.04(0.43) -0.02(-0.18) 0.04(0.50) 0.04(0.48)

Large 0.07(0.84) 0.08(0.92) 0.05(0.63) 0.06(0.71) 0.07(0.81)

Organizational identification 0.22**(4.73) 0.21**(4.28)

Adj.R2 0.20 0.16 0.18 0.24 0.19

F 18.71** 14.81** 16.24** 20.04** 15.83**

Notes: Standard coefficients are reported, with t values in parentheses; n=432. *p<0.05, **p<0.01. The variance inflation factors (VIF) for all variables in the models were below 3. OI=organizational identification, IP=in-role performance, EP=extra-role performance. Homoscedasticity was supported by the plot of standardized residuals against predicted values.

Additionally, given the large sample size of this study, the Central Limit Theorem ensures that the sampling distributions of the estimated model parameters (such as β1 and β2) will approach normality, even if the error terms are not perfectly normally distributed. Therefore, no additional explanation regarding normality tests has been included in the revisions.

3. Results – Clarity and Integration with Hypotheses

Results are presented in a logical order, but in some cases the link between numerical outputs and the corresponding hypotheses could be made more explicit in the text. This would strengthen the narrative flow and help the reader follow how the evidence addresses each hypothesis.

Additionally, while figures are generally clear, in Figure 1 there is a typographical error (perceived not perceive CSR) and Figure 3 have axes or legends that are difficult to interpret without referring to the text. Also captions should be more self-contained.

My reply：

I have performed major revisions on the "Results" section to enhance its clarity and make it self-contained. For instance, key statistical measures have been incorporated into this section, as illustrated below (see text in blue)：

Drawing on a sample of Chinese manufacturing employees, this study examined their perceived CSR, organizational identification, work performance, and altruistic values, with a focus on the relationships among these variables. The empirical results support the theoretical framework (Figure 1) and all three research hypotheses.

The current research confirms that manufacturing workers perceive a high level of CSR (M=4.05, SD = 0.69; Table 1), consistent with a previous study[3], which indicated that Chinese organizations prioritize social responsibility practices. The employees exhibit strong organizational identification (M=4.11, SD=0.68; Table 1), indicating that they have a deep connection with their respective organizations. Furthermore, supervisory ratings indicate moderate levels of in-role performance (M=3.34, SD=1.02; Table 1), and high levels of extra-role performance (M=3.79, SD=0.93; Table 1) among employees, suggesting that frontline manufacturing workers generally demonstrate significant work effort. Lastly, employees report low levels of altruistic values (M=2.45, SD=0.75; Table 1), indicating a limited concern for the well-being or welfare of others.

This study confirms a moderate positive link between perceived CSR and both in-role (r=0.45) and extra-role (r=0.41, Table 2) performance, consistent with prior research[53]. For instance, a meta-analysis confirmed that the correlation coefficient between perceived CSR and employee performance generally ranges from 0.33 to 0.45 [54]. Additionally, this study revealed that employees' perceived CSR has a moderately positive correlation with organizational identification (r=0.42, Table 2), consistent with previous findings [1, 55]. For example, an earlier study reported a Pearson correlation of 0.57 between perceived CSR and organizational identification[55]. Additionally, organizational identification is weakly and positively related to both in-role (r=0.37) and extra-role (r=0.34, Table 2) performance, consistent with the findings in a prior study [1].

Regression analysis, controlling for gender, education, tenure, and firm size, showed that perceived CSR had a significant positive effect on both in-role (β = 0.44, p <0 .01) and extra-role performance (β = 0.40, p < 0.01，Table 2), thus supporting Hypothesis 1, which posits that perceived CSR is positively related to employee performance. This conclusion aligns closely with results from previous studies. For instance, one study demonstrated that employees in high-CSR environments outperform their peers[10], while another found that employees' perceived CSR enhance their organizational citizenship behavior [56]. This result not only echoes the findings from studies conducted in Western cultures [57, 58], but also shows that CSR has a positive relationship with employee performance within the context of Chinese culture.

The regression analysis also indicated that perceived CSR has a significant effect on organizational identification (β = 0.39, p < 0.01; Model 2 in Table 2). Furthermore, when both organizational identification and perceived CSR were included, organizational identification demonstrated positive effects on both in-role (β = 0.22, p < 0.01; Model 3a in Table 2) and extra-role performance (β = 0.21, p < 0.01; Model 3b in Table 2). Therefore, Hypothesis 2 is strongly supported, positing that organizational identification mediates the relationship between perceived CSR and employee performance. This finding is consistent with the results of previous research. For example, a prior study demonstrated that organizational identification mediates the relationships between perceived CSR and turnover intention, in-role job performance, and extra-role performance, as indicated by helping behavior [1, 59]. Additionally, a study conducted in China found that organizational identification plays a crucial role in the relationship between employee-perceived CSR and organizational citizenship behavior[6].

⋯⋯

Regression results indicated a significant interaction between perceived CSR and altruistic values on organizational identification (β = -0.10, p < 0.05; Table 3), supporting Hypothesis 3 that altruistic values moderate the positive relationship between CSR and identification. The negative moderating effect suggests that, for employees with high levels of altruism, the positive association between perceived CSR and organizational identification is attenuated compared to those with lower altruism. Nevertheless, employees with high altruism tend to exhibit stronger organizational identification than those with low altruism (Table 4 and Figure 3). This finding aligns with a prior study showing that employees' altruistic beliefs moderate the po

---

## [Decision Letter · Decision Letter 1]

6 Nov 2025

PONE-D-25-38655R1

Corporate Social Responsibility and Employee Performance in China's Manufacturing Sector: Exploring the Roles of Altruistic Values and Organizational Identification

PLOS ONE

Dear Dr. Ruanggoon,

Thank you for submitting your manuscript to PLOS ONE. After careful consideration, we feel that it has merit but does not fully meet PLOS ONE’s publication criteria as it currently stands. Therefore, we invite you to submit a revised version of the manuscript that addresses the points raised during the review process.

We look forward to receiving your revised manuscript.

Kind regards,

Federico Zilia

Academic Editor

PLOS ONE

**Journal Requirements:**

**Additional Editor Comments:**

After careful consideration, I have decided to request a Minor Revision before considering the acceptance. Please address the following specific points:

Expand the Introduction with a brief summary of the study’s main findings and structure.In the Limitations section, explicitly note that the sample was non-probabilistic and discuss the implications for generalizability.Clarify that Harman’s test was conducted as an ex-post check for common method bias (see Podsakoff et al., 2003).Conduct a final proofreading to ensure smooth, concise language.

Once these minor issues are corrected, your manuscript should be ready for publication.

Reviewers' comments:

Reviewer's Responses to Questions

**Comments to the Author**

1. If the authors have adequately addressed your comments raised in a previous round of review and you feel that this manuscript is now acceptable for publication, you may indicate that here to bypass the “Comments to the Author” section, enter your conflict of interest statement in the “Confidential to Editor” section, and submit your "Accept" recommendation.

Reviewer #1: All comments have been addressed

Reviewer #2: (No Response)

2. Is the manuscript technically sound, and do the data support the conclusions?

Reviewer #1: Yes

Reviewer #2: Partly

3. Has the statistical analysis been performed appropriately and rigorously?

Reviewer #1: Yes

Reviewer #2: No

4. Have the authors made all data underlying the findings in their manuscript fully available?

Reviewer #1: Yes

Reviewer #2: Yes

5. Is the manuscript presented in an intelligible fashion and written in standard English?

Reviewer #1: Yes

Reviewer #2: Yes

6. Review Comments to the Author

Reviewer #1: Dear Author/s,

I have reviewed the revised version of the manuscript entitled "Corporate Social Responsibility and Employee Performance in China's Manufacturing Sector: Exploring the Roles of Altruistic Values and Organizational Identification" (PONE-D-25-38655_R1). The authors have made a genuine effort to address the reviewers’ comments and have substantially improved the clarity and overall quality of the paper.

In particular, the introduction has been streamlined, the methodological section expanded with additional details on variable selection and statistical checks, and the results are now better aligned with the stated hypotheses. Minor issues in figures and captions have also been properly corrected.

However, I would still recommend a minor revision before final acceptance. The Discussion and Conclusions sections remain rather concise and would benefit from deeper engagement with the findings. Specifically:

- A more explicit consideration of alternative explanations and the broader theoretical implications would strengthen the discussion.

- The conclusion could more clearly articulate the study’s limitations (I suggest to move the limitation section in the conclusions) and outline possible directions for future research.

Finally, a careful language proofreading is advised to smooth a few long or complex sentences.

Overall, the manuscript is solid, original, and ready for publication after these minor adjustments.

Sincerely,

Reviewer #2: INTRODUCTION

The introduction positions the analysis consistently within the relevant literature as in the previous version. Although there has been an effort to clarify the theoretical framework in which the analysis is situated, my previous comments on this section have not been fully addressed. I restate them here: “some adjustments are necessary to make the manuscript more intelligible. The introduction should cover both the theoretical foundations and the related empirical contributions. In addition, it should provide a brief explanation of the objectives of the study and the overall structure of the paper. Ideally, one should be able to grasp the essence of the work by reading only the introduction and the conclusions. This is not the case here and I would therefore recommend including at least a summary of the main findings as well as a clearer description of the paper’s structure.”

Other minor issues:

- Typo: “For example, an study of multinational financial corporations revealed”

- I would avoid mentioning literature that highlights how organizational identification moderates the impact of perceived CSR on task performance and helping behavior in manufacturing environments when discussing its mediating effect (p. 4).

METHODS

My previous comments have been only partially addressed.

- “How were the enterprises selected? Without this information the results may not be generalizable to the target population. This clarification is essential in order to properly assess the validity of the manuscript’s findings.” Purposive sampling is not a probabilistic method, therefore it would be important to explain how the enterprises were selected and to acknowledge the limitations of this sampling approach. The results and conclusions should be discussed in light of these limitations. Accordingly, the “Limitations” section should be expanded.

Minor points:

- Include a reference to the appendix for the listing of measurement instruments (a footnote would also suffice).

- The expression “deprived from a previous study” should be revised, as it appears to be incorrect.

- The subsection “Data Analysis” is too brief. Key information should be presented here rather than scattered across the results section, otherwise the subsection loses its purpose.

RESEARCH RESULTS

As with the previous sections, my earlier comments have been only partially addressed.

Major remarks:

- Harman’s single-factor test is an ex-post statistical check, not a method to anticipate common method bias. Claiming otherwise is simply incorrect.

- Given that the outcome variables were evaluated by supervisors who did not receive rigorous training (your own words), it would be useful to provide a robustness check controlling for the individual supervisor.

- Regarding the CFA, it is necessary to report the total variance explained by the construct. I therefore recommend following the guidance in Podsakoff et al. (2003).

Minor points:

- As a side note: CSR and OI are latent, complex, and unobservable constructs that the authors are attempting to measure. The current method does not explicitly account for how well the items capture the underlying latent variable, nor does it handle measurement error as SEM would. Furthermore, with this approach, important issues such as differential item functioning cannot be considered when estimating latent constructs. Additionally, SEM would allow controlling for demographic variables as in a standard regression framework. I therefore suggest exploring SEM-based approaches in the future.

- On page 13 (text in red), Model 2 and Model 3 appear to be described identically, although Table 2 clearly indicates that Model 2 is actually a regression of OI on CSR. This should be explained better.

- The explanation of the partial mediating effect of OI is unsatisfactory. It would be preferable to indicate what happens to the relevant coefficients when OI is included in Model 3.

- I would underline the fact that the interaction term in table 3 is significant at 5% level.

DISCUSSION, IMPLICATIONS, LIMITATIONS AND CONCLUSIONS

These sections should be reconsidered in light of the requested revisions. At present, the claims are not, in the reviewer’s view, sufficiently well supported.

Bibliography

Podsakoff, P. M., MacKenzie, S. B., Lee, J. Y., & Podsakoff, N. P. (2003). Common method biases in behavioral research: a critical review of the literature and recommended remedies. Journal of applied psychology, 88(5), 879.

7. PLOS authors have the option to publish the peer review history of their article (what does this mean?). If published, this will include your full peer review and any attached files.

Reviewer #1: No

Reviewer #2: No

---

## [Author Response · Author response to Decision Letter 2]

12 Nov 2025

Dear Editors and Reviewers,

Thank you very much for your comments! I have carefully read your evaluations and suggestions, thoroughly reviewed the manuscript, and made revisions one by one in accordance with your requirements. Please feel free to inform me if there are any inappropriate parts, and I will revise them diligently again. Thank you so much!

To editor

Editor’s comment 1: Expand the Introduction with a brief summary of the study’s main findings and structure.

My reply:

I have added a brief summary of the study’s main findings and structure in the "Introduction" section as follows (see the content in blue).

This study demonstrates that the relationship between perceived CSR and employee performance is partially mediated by organizational identification, and that altruistic values negatively moderate the link between perceived CSR and organizational identification. It enriches SIT within the Chinese cultural context and contributes to improving management practices in Chinese enterprises. The remainder of the study is organized as follows: (1) Methods (sampling and measures); (2) Research results (mediation and moderation analyses); (3) Discussion; (4) Implications; (5) Limitations; (6) Conclusions; and (7) Appendix (containing the measurement scales).

Editor’s comment 2: In the Limitations section, explicitly note that the sample was non-probabilistic and discuss the implications for generalizability.

My reply:

To highlight the limitations of non-probabilistic sampling, I have added the following content in the "Limitations" section (see the content in blue).

Finally, this study employed a non-probabilistic sampling method, drawing its sample exclusively from manufacturing firms in Sichuan Province. This approach may thereby constrain the generalizability of the findings to other regions or industries. To enhance the external validity of future research, it would be beneficial to employ probability sampling techniques across more diverse geographical contexts..

Editor’s comment 3: Clarify that Harman’s test was conducted as an ex-post check for common method bias (see Podsakoff et al., 2003).

My reply:

I have added the following content in the "Common Method Variance "section (see the content in blue).

First, Harman's single-factor test was conducted as an ex-post check[45].

I have also added the literature you provided i in the references (see the content in blue):

45. Podsakoff PM, MacKenzie SB, Lee J-Y, Podsakoff NPJJoap. Common method biases in behavioral research: a critical review of the literature and recommended remedies. Journal of applied psychology. 2003;88(5):879.

Editor’s comment 4: Conduct a final proofreading to ensure smooth, concise language.

My reply:

I have carefully read through the entire manuscript and revised every detail that required modification. I firmly believe that the quality of my manuscript has been greatly improved and now meets the publication standards of PLOS ONE.

Finally, I have carefully reviewed all the literature and found no retracted papers. All the references are from truly reliable sources.

To reviewer 1

Comment 1: A more explicit consideration of alternative explanations and the broader theoretical implications would strengthen the discussion.

My reply:

I have appropriately added content related to alternative explanations in the "Discussion" section (see the content in blue):

For example, one study found that organizational trust and job satisfaction can mediate the relationship between perceived CSR and employee performance[65]. Another study demonstrated that work meaningfulness and organizational pride sequentially mediate the link between perceived CSR and job performance among Chinese employees[66]. In addition, another potential explanatory mechanism deserves consideration. Within China's cultural context, where authoritarian norms are prevalent [67], employees' work performance and CSR compliance may result more from deference and loyalty to management than from genuine endorsement of CSR or organizational identification. In summary, beyond the mechanism of organizational identification grounded in SIT, other mechanisms may also partially account for the relationship between perceived CSR and employee performance.

Comment 2: The conclusion could more clearly articulate the study’s limitations (I suggest to move the limitation section in the conclusions) and outline possible directions for future research.

My reply:

I have carefully revised the "Limitations" section and clearly stated the potential future research directions (see the content in blue).

This study has three notable limitations that warrant attention. First, it employs cross-sectional data, which allows only correlational, not causal, conclusions. Future research should incorporate longitudinal follow-ups with participants to provide more robust evidence supporting the relationships among perceived CSR, organizational identification, and employee performance. Second, the present study examines the associations between perceived CSR and employee performance using an individual-level model, thereby overlooking potential organization-level influences. Future research would benefit from exploring employees' responses to CSR within a multilevel framework. Finally, this study employed a non-probabilistic sampling method, drawing its sample exclusively from manufacturing firms in Sichuan Province. This approach may thereby constrain the generalizability of the findings to other regions or industries. To enhance the external validity of future research, it would be beneficial to employ probability sampling techniques across more diverse geographical contexts.

Comment 3: Finally, a careful language proofreading is advised to smooth a few long or complex sentences.

My reply:

I have carefully read every sentence of the manuscript and conducted a careful language proofreading.

To reviewer 2

Comment 1: The introduction should cover both the theoretical foundations and the related empirical contributions. In addition, it should provide a brief explanation of the objectives of the study and the overall structure of the paper. Ideally, one should be able to grasp the essence of the work by reading only the introduction and the conclusions. This is not the case here and I would therefore recommend including at least a summary of the main findings as well as a clearer description of the paper’s structure.”

My reply:

I have carefully read every sentence of the manuscript and conducted a careful language proofreading. Many minor revisions cannot be listed one by one here. In short, I have made every effort to improve the quality of the manuscript.

Meanwhile, I have provided a brief explanation of the study’s objectives and the overall structure of the paper; please refer to the following content (in blue):

This study demonstrates that the relationship between perceived CSR and employee performance is partially mediated by organizational identification, and that altruistic values negatively moderate the link between perceived CSR and organizational identification. It enriches SIT within the Chinese cultural context and contributes to improving management practices in Chinese enterprises. The remainder of the study is organized as follows: (1) Methods (sampling and measures); (2) Research results (mediation and moderation analyses); (3) Discussion; (4) Implications; (5) Limitations; (6) Conclusions; and (7) Appendix (containing the measurement scales).

Comment 2: Typo: “For example, an study of multinational financial corporations revealed”

My reply:

I have corrected this minor mistake, as shown below (in blue font):

For example, an empirical study of multinational financial corporations revealed that community-oriented CSR positively influenced employees' in-role performance,

Comment 3: I would avoid mentioning literature that highlights how organizational identification moderates the impact of perceived CSR on task performance and helping behavior in manufacturing environments when discussing its mediating effect (p. 4).

My reply:

I have deleted this paragraph and added the corresponding content as follows (see blue font):

Quantitative studies based on SIT revealed that organizational identification could mediate the relationship between CSR and employee outcomes. A study in Western cultural contexts revealed that perceived CSR enhanced organizational identification, which in turn improved employee adjustment and job performance[24]. Organizational identification and job satisfaction were found to serially mediate the link between CSR and work performance among Korean employees [25], while organizational identification and commitment channeled CSR's effects on performance and turnover intention in Saudi workers[26]. Consistent with these findings, a Chinese study found that organizational identification mediated the relationship between perceived CSR and employee behaviors, including turnover intention, helping behavior, and in-role performance[1].

Comment 4: Purposive sampling is not a probabilistic method, therefore it would be important to explain how the enterprises were selected and to acknowledge the limitations of this sampling approach. The results and conclusions should be discussed in light of these limitations.

My reply:

First, the following content (see blue font) has been added to the "Sampling Procedure" section to clarify the specific details of the sampling:

The study employed a purposive sampling approach to identify eligible enterprises based on predefined criteria: (1) operation in the manufacturing sector, (2) documented implementation of CSR initiatives, and (3) expressed willingness to participate in the survey. Subsequently, a convenience sampling method was used to recruit accessible frontline supervisors—each required to have at least ten direct subordinates and agreed to cooperate—along with their respective teams within these organizations. The sample comprised three state-owned and four privately owned firms from diverse sectors, including machinery, food production, household appliances, and apparel. The employees surveyed were exclusively front-line workers responsible for production tasks, quality control, sales, marketing, and related activities. We contacted the direct supervisors of these frontline employees across the companies and requested their assistance in facilitating the investigation. This approach ensured the confidentiality of the participants[37].

Second, I have revised the corresponding content in the "Limitations" section to highlight the impact of non-probability sampling on the research results (see blue font):

Finally, this study employed a non-probabilistic sampling method, drawing its sample exclusively from manufacturing firms in Sichuan Province. This approach may thereby constrain the generalizability of the findings to other regions or industries. To enhance the external validity of future research, it would be beneficial to employ probability sampling techniques across more diverse geographical context..

Comment 5: Include a reference to the appendix for the listing of measurement instruments (a footnote would also suffice).

My reply:

I have added "see Appendix 1" to the "Measurement Instruments" section, as shown in the following content (see blue font):

The measure of CSR is adapted from a prior study [38]. It encompasses six key items (see Appendix 1):

A 5-item scale was used to measure organizational identification [39] (see Appendix 1).

Employee performance was measured using a six-item scale adapted from two previous studies [43, 44] (see Appendix 1).

Comment 6: The expression “deprived from a previous study” should be revised, as it appears to be incorrect.

My reply:

I have revised this improperly expressed part, as shown below (in blue font): One of the items was: donate clothes or goods to people in need. These items were adapted from previous studies[40-42].

Comment 7: The subsection “Data Analysis” is too brief. Key information should be presented here rather than scattered across the results section, otherwise the subsection loses its purpose.

My reply:

I have made appropriate revisions to the description of data processing to make it clearer (see blue font):

First, Harman's single-factor test was conducted as an ex-post check[45]. It revealed four factors with eigenvalues exceeding 1, with the first factor explaining 32.72% of the total variance—below the 40% threshold—indicating no significant common method bias in the survey data[46]. Second, confirmatory factor analysis for a single latent factor demonstrated poor model fit (RMSEA = 0.13, CFI = 0.56, TLI = 0.51, SRMR = 0.11). The common factor accounted for only 28.57% of the total variance on average (R² range: 0.13–0.44), which is well below the 50% threshold, thereby confirming the absence of common method bias.

Comment 8: Harman’s single-factor test is an ex-post statistical check, not a method to anticipate common method bias. Claiming otherwise is simply incorrect.

My reply:

I have added a sentence to the "Common Method Variance" section to emphasize "an ex-post statistical check" (see blue font).

First, Harman's single-factor test was conducted as an ex-post check[45]. It revealed four factors with eigenvalues exceeding 1, with the first factor explaining 32.72% of the total variance—below the 40% threshold—indicating no significant common method bias in the survey data[46]. Second, confirmatory factor analysis for a single latent factor demonstrated poor model fit (RMSEA = 0.13, CFI = 0.56, TLI = 0.51, SRMR = 0.11). The common factor accounted for only 28.57% of the total variance on average (R² range: 0.13–0.44), which is well below the 50% threshold, thereby confirming the absence of common method bias.

In addition, the following text has been added to the "Measurement Instruments" section (see the blue font).：

To minimize potential common method bias, all measurement items were carefully developed to be clear, simple, and concise, avoiding ambiguous or double-barreled statements. In addition, items for all constructs used specific and unambiguous wording to reduce context-induced mood effects and were further randomized in the questionnaire[45, 46].

Comment 9: Regarding the CFA, it is necessary to report the total variance explained by the construct. I therefore recommend following the guidance in Podsakoff et al. (2003).

My reply:

Finally, I have added a sentence to the "Common Method Variance" section to present the total variance explained by the construct (see blue font).

Second, confirmatory factor analysis for a single latent factor demonstrated poor model fit (RMSEA = 0.13, CFI = 0.56, TLI = 0.51, SRMR = 0.11). The common factor accounted for only 28.57% of the total variance on average (R² range: 0.13–0.44), which is well below the 50% threshold, thereby confirming the absence of common method bias.

Comment 10: On page 13 (text in red), Model 2 and Model 3 appear to be described identically, although Table 2 clearly indicates that Model 2 is actually a regression of OI on CSR. This should be explained better.

My reply:

I have carefully revised this part of the content to make the expression clearer (see the content in blue font):

Secondly, Model 2 revealed that perceived CSR had a strong predictive effect on organizational identification (β= 0.39, p< 0.01).

Thirdly, when both organizational identification and perceived CSR were included in the model, organizational identification significantly predicted both in-role (β = 0.22, p <0.01) and extra-role performance (β = 0.21, p <0.01). Concurrently, the predictive effects of perceived CSR decreased but remained significant, from β = 0.44 to 0.36 for in-role performance and from β = 0.40 to 0.32 for extra-role performance (p <0.01 for both; Model 3a and 3b). Therefore, organizational identification serves as a partial mediator in the relationship between perceived CSR and employee performance, providing support for Hypothesis 2.

Comment 11: The explanation of the partial mediating effect of OI is unsatisfactory. It would be preferable to indicate what happens to the relevant coefficients w

---

## [Decision Letter · Decision Letter 2]

8 Dec 2025

Corporate Social Responsibility and Employee Performance in China's Manufacturing Sector: Exploring the Roles of Altruistic Values and Organizational Identification

PONE-D-25-38655R2

Dear Dr. Jirapong Ruanggoon,

We’re pleased to inform you that your manuscript has been judged scientifically suitable for publication and will be formally accepted for publication once it meets all outstanding technical requirements.

Kind regards,

Federico Zilia

Academic Editor

PLOS One

Additional Editor Comments (optional):

Reviewers' comments:

Reviewer's Responses to Questions

**Comments to the Author**

1. If the authors have adequately addressed your comments raised in a previous round of review and you feel that this manuscript is now acceptable for publication, you may indicate that here to bypass the “Comments to the Author” section, enter your conflict of interest statement in the “Confidential to Editor” section, and submit your "Accept" recommendation.

Reviewer #1: All comments have been addressed

Reviewer #2: All comments have been addressed

2. Is the manuscript technically sound, and do the data support the conclusions?

Reviewer #1: Yes

Reviewer #2: Yes

3. Has the statistical analysis been performed appropriately and rigorously?

Reviewer #1: Yes

Reviewer #2: Yes

4. Have the authors made all data underlying the findings in their manuscript fully available?

Reviewer #1: Yes

Reviewer #2: Yes

5. Is the manuscript presented in an intelligible fashion and written in standard English?

Reviewer #1: Yes

Reviewer #2: Yes

6. Review Comments to the Author

Reviewer #1: I have carefully reviewed the revised manuscript PONE-D-25-38655_R2. The authors have thoroughly addressed the issues raised in the previous round and made substantial improvements to both structure and content.

The introduction now clearly presents the research objectives, theoretical foundations, and the structure of the paper. The methodology section has been strengthened, with full details about the purposive and convenience sampling procedures, control variables, and data-analysis steps (including clear treatment of Harman’s test as an ex-post check for common-method bias). The results and tables are consistent and properly referenced.

Most importantly, the discussion and conclusions have been expanded as recommended. The authors now explicitly acknowledge alternative explanatory mechanisms and theoretical implications and provide a thoughtful account of limitations and directions for future research. These additions considerably improve interpretive depth. The language and style have also been carefully proofread; only very minor grammatical polishing might still be advisable at the copy-editing stage.

Reviewer #2: (No Response)

7. PLOS authors have the option to publish the peer review history of their article (what does this mean?). If published, this will include your full peer review and any attached files.

Reviewer #1: No

Reviewer #2: No

---

## [Editor Report · Acceptance letter]

PONE-D-25-38655R2

PLOS One

Dear Dr. Ruanggoon,

I'm pleased to inform you that your manuscript has been deemed suitable for publication in PLOS One. Congratulations! Your manuscript is now being handed over to our production team.

Kind regards,

on behalf of

Dr. Federico Zilia

Academic Editor

PLOS One